

# Extensive analysis of native and non-native *Centaurea solstitialis* L. populations across the world shows no traces of polyploidization

Ramona-Elena Irimia[1,2], Daniel Montesinos[1], Özkan Eren[3], Christopher J. Lortie[4], Kristine French[5], Lohengrin A. Cavieres[6,7], Gastón J. Sotes[6,7], José L. Hierro[8], Andreia Jorge[1] and João Loureiro[1]

[1] Centre for Functional Ecology, Department of Life Sciences, University of Coimbra, Coimbra, Portugal
[2] National Institute of Research and Development for Biological Sciences, Stejarul Research Centre for Biological Sciences, Piatra Neamt, Romania
[3] Adnan Menderes Üniversitesi, Fen-Edebiyat Fakültesi, Biyoloji Bölümü, Aydın, Turkey
[4] Department of Biology, York University, Toronto, Canada
[5] School of Biological Sciences, University of Wollongong, Wollongong, Australia
[6] Departamento de Botánica, Facultad de Ciencias Naturales y Oceanográficas, Universidad de Concepción, Concepción, Chile
[7] Instituto de Ecología y Biodiversidad (IEB), Santiago, Chile
[8] Instituto de Ciencias de La Tierra y Ambientales de la Pampa, Consejo Nacional de Investigaciones Científicas y Técnicas (INCITAP-CONICET) and Facultad de Ciencias Exactas y Naturales, Universidad Nacional de La Pampa, Santa Rosa, Argentina

Corresponding author
Ramona-Elena Irimia,
ramonaeirimia@gmail.com

## ABSTRACT

*Centaurea solstitialis* L. (yellow starthistle, Asteraceae) is a Eurasian native plant introduced as an exotic into North and South America, and Australia, where it is regarded as a noxious invasive. Changes in ploidy level have been found to be responsible for numerous plant biological invasions, as they are involved in trait shifts critical to invasive success, like increased growth rate and biomass, longer life-span, or polycarpy. *C. solstitialis* had been reported to be diploid ($2n = 2x = 16$ chromosomes), however, actual data are scarce and sometimes contradictory. We determined for the first time the absolute nuclear DNA content by flow cytometry and estimated ploidy level in 52 natural populations of *C. solstitialis* across its native and non-native ranges, around the world. All the *C. solstitialis* populations screened were found to be homogeneously diploid (average 2C value of 1.72 pg, SD = ±0.06 pg), with no significant variation in DNA content between invasive and non-invasive genotypes. We did not find any meaningful difference among the extensive number of native and non-native *C. solstitialis* populations sampled around the globe, indicating that the species invasive success is not due to changes in genome size or ploidy level.

## INTRODUCTION

Changes in ploidy level have been reported to be important for the invasive success of some plants species (*Te Beest et al., 2011*), by altering morphological, physiological and ecological parameters which can confer hybrid vigor, stress resistance, competitive advantages, or increased phenotypic plasticity, like in the case of the North American tetraploids of *Centaurea stoebe* L. (*Hahn, Buckley & Müller-Schärer, 2012*). Additionally, there are a series of associated "genome size constrained traits", related mostly to reproduction and dispersal, which dictate the ecological niche a species can access (*Te Beest et al., 2011*). In contrast, several studies support the hypothesis that a smaller genome can contribute to some species invasive potential by boosting early plant growth and enhancing competitive ability (*Bennett, Leitch & Hanson, 1998*; *Grotkopp et al., 2004*; *Beaulieu et al., 2007*; *Lavergne, Muenke & Molofsky, 2010*; *Suda et al., 2015*). For instance, *Phalaris arundinacea* L. (reed canary grass, Poaceae) in the USA underwent a quick and significant reduction in genome size compared to the native European genotype, which was correlated with some advantageous phenotypic effects and enhanced aggressiveness (*Lavergne, Muenke & Molofsky, 2010*). A list comparing the ploidy level of 128 worst invasive plant species worldwide, was recently made available by *Te Beest et al. (2011)*, indicating that a quarter of them possess at least two different ploidy levels. An interesting example is *C. stoebe* (spotted knapweed) which occurs both as a diploid and tetraploid, with only the latter cytotype becoming invasive in the Western parts of the USA (*Mráz et al., 2011*). However, for many invasive species, ploidy levels and genome size are unknown or have not been thoroughly investigated.

*Centaurea* L. is one of the most species rich genera in the Asteraceae (*Bremer, 1994*). Numerous *Centaurea* species have been introduced into new non-native regions, where many of them have become invasive. For instance, the US Federal Noxious Weeds list (*USDA, NRCS, The PLANTS Database, 2017*), includes no fewer than 13 taxa, but ploidy level for many of these is unknown or uncertain. In particular, *C. solstitialis* is a Eurasian native annual herb which was introduced into the Americas and Australia during the last two centuries (*Barker et al., 2017*) and became an impactful invader in the former case. In the invaded ranges, *C. solstitialis* forms dense stands that displace native plants species and reduce considerably livestock grazing capacity and forage value (*Eagle et al., 2007*). It alters ecosystem functions by depleting soil water and nutrients through an extensive root system (*DiTomaso, 2000*), and can cause a neurological disorder in horses similar to human Parkinson (*Chang et al., 2011*). As an economically important plant, the species has been the subject of intensive research, and significant differentiation between native and non-native ranges have been reported for plant size (*Eriksen et al., 2012*; *Graebner, Callaway & Montesinos, 2012*; *García et al., 2013*; *Dlugosch et al., 2015*), growth rates (*Graebner, Callaway & Montesinos, 2012*), germination (*Hierro et al., 2009*), competitive ability (*Montesinos & Callaway, 2017*), and reproduction (*Montesinos, Santiago & Callaway, 2012*), among others. Such changes suggest diverging local adaptation occurring among native and non-native ranges, and hypothetical changes in genome

size and ploidy level could be potentially responsible for at least some of the observed trait-shifts.

Until now, only three genome size estimates were available in the literature for *C. solstitialis*: two from the native range (Bulgaria: 1.74 pg/2C, one accession, in *Bancheva & Greilhuber, 2006*; and Croatia: 1.95 pg/2C, five accessions, in *Carev et al., 2017*) and another from an invasive population in western USA: 1.66 pg/2C, thirty accessions (*Miskella, 2014*). Based on these few studies, *C. solstitialis* had been reported to be diploid (*Dlugosch et al., 2013*; *Rice et al., 2015*) with $2n = 2x = 16$ chromosomes. However, records of $2n = 2x = 18$ chromosomes were published more than 30 years ago from the native range of Bulgaria (*Jasiewicz & Mizianty, 1975*; *Kuzmanov, Jurukova-Grancarova & Georgieva, 1990*) and recently from one accession from Sicily and the other one from Sardinia (*Widmer et al., 2007*). Furthermore, *Inceer, Hayirlioglu-Ayaz & Ozcan (2007)* reported tetraploids in seeds (single accession) sampled in northern Turkey, but none of those observations, made in only a handful of individuals, have been confirmed since then. Consequently, it was still unclear whether ploidy could have played a role in at least some of the *C. solstitialis* invaded ranges. To fill this knowledge gap for such an important species, we aimed to thoroughly sample and assess *C. solstitialis* ploidy level and genome size in a representative number of populations from around the world, including native Turkey, the ancestral origin of the species; native Spain, the main source of American populations; and all the known non-native regions represented by Argentina, Chile, USA and Australia.

## METHODS

### Seed collection

A total of 477 accessions from 52 natural populations (Table S1) of *C. solstitialis* were investigated in this study, for genome size and ploidy level assessment. Within the native area, we sampled ten populations from Turkey, near the Caucasus region, where high genetic diversity has been detected, and is regarded as the site of origin of the species (*Wagenitz, 1955*; *Gerlach, 1997a*; *Uygur et al., 2004*; *Dlugosch et al., 2013*; *Eriksen et al., 2014*), and ten populations from Spain, considered as the primary source of seeds to have colonized Chile and Argentina (*Hijano & Basigalup, 1995*; *Eriksen et al., 2012*; *Eriksen et al., 2014*; *Dlugosch et al., 2013*; *Barker et al., 2017*) in the nineteenth century (*Gerlach, 1997b*). For the non-native regions, we included ten populations from Argentina and California, eight from Australia and four from Chile. Seeds were extracted from mature flower heads collected in the wild from ten individuals per population between 2009 and 2014. Ten seeds from each individual were germinated in plant growing trays, under common greenhouse conditions, in early spring 2016 at the Botanical Garden of the University of Coimbra, Portugal.

### Flow cytometry

Young and intact leaves of 4–6 weeks-old plants were sampled and screened by flow cytometry. Since analyses were based on leaves of small plants, which were destroyed by leaf sampling, no voucher specimens could be collected. Nuclei were isolated following
the chopping method of *Galbraith et al. (1983)*. Briefly, about 1 cm$^2$ of leaf tissue was co-chopped with a razor blade together with the same amount of reference standard (*Raphanus sativus* L. 'Saxa', 2C = 1.11 pg, *Doležel, Sgorbati & Lucretti, 1992*) in 1 mL of woody plant buffer (WPB): 0.2 M Tris×HCl, 4 mM MgCl$_2$×6H$_2$O, 2 mM EDTA Na$_2$×2H$_2$O, 86 mM NaCl, 10 mM sodium metabisulfite, 1% polyvinylpyrrolidone (PVP-10) (w/v) and 1% Triton X-100 (v/v), with pH of the buffer adjusted to 7.5 (*Loureiro et al., 2007*). The resulting homogenate was filtered through a 50 μm nylon filter into a sample tube to remove large debris. Nuclei were stained with 50 mg/mL propidium iodide (PI; Fluka, Buchs, Switzerland), and 50 mg/ml of RNAse (Fluka, Buchs, Switzerland) was added to prevent the staining of double stranded RNA. Samples were kept at room temperature and analyzed immediately on a Partec CyFlow Space flow cytometer (Partec GmbH, Görlitz, Germany) equipped with a 532 nm green solid-state laser, operating at 30 mW.

### Data collection and analysis

Results were acquired using Partec FloMax software (v2.4d) (Partec GmbH, Münster, Germany) in the form of six graphics: fluorescence pulse integral in linear scale (FL); forward light scatter (FS) *vs*. side light scatter (SS), both in logarithmic (log) scale; FL *vs*. time; FL *vs*. fluorescence pulse height; FL *vs*. FS in log scale and FL *vs*. SS in log scale. Mean fluorescence values and coefficient of variation (CV value) of the fluorescence of both sample and standard were obtained for at least 1,300 nuclei in each G$_1$ peak, whenever possible. Samples with CV values above 5% were discarded, prepared and ran again. At least three individuals from every population were used to estimate genome size (Table S2), in different days, to account for the variation generated by the flow cytometer. The remaining individuals were analyzed in pool (three or four individuals) to determine ploidy level (Table S2), only. The absolute DNA content of a sample was calculated based on the following formula: 2C nuclear DNA content of the sample = (sample G$_1$ peak mean)/(standard G$_1$ peak mean) × 2C DNA content of standard. Descriptive statistics were calculated for genome size data (mean, standard deviation of the mean, standard error, coefficient of variation and minimum and maximum values) using Microsoft Excel 2016. Differences in average genome size values among regions were assessed by means of Linear Mixed-Effect Models with the formulation of *Laird & Ware (1982)*, with a region as fixed factor and population within region as a random nested factor, in R-3.2.0 (*R Development Core Team, 2010*). Data was plotted in BoxPlotR (*Spitzer et al., 2014*).

## RESULTS

Analysis of fresh leaf tissue sampled from seedlings germinated from wild seeds of individuals from 52 populations from Turkey, Spain, Argentina, Chile, USA and Australia (Table S1), showed no significant differences in genome size ($F_{5,44} = 0.58$; $p = 0.716$) among regions (Fig. 1). All individuals ($N = 477$) were found to be diploid, presumably with $2n = 16$ chromosomes. Average genome size ranged from 1.70 pg/2C (SD = 0.06 pg) in Australia and Spain (SD = 0.06 pg) to 1.71 pg/2C (SD = 0.06 pg) in Chile, 1.72 pg/2C (SD = 0.06 pg) in Argentina and California (SD = 0.07 pg) and 1.73 pg/2C (SD = 0.07 pg) in Turkey (Table 1).

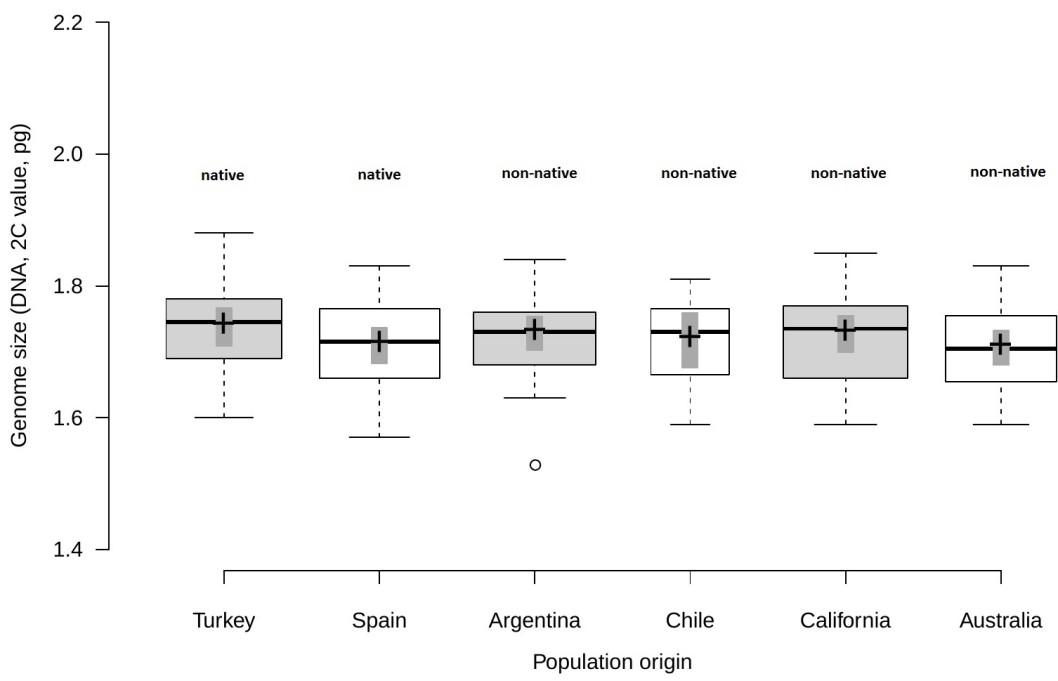

**Figure 1 Comparison of genome size among native and non-native genotypes of *Centaurea solstitialis*.**
Black center lines represent the medians, crosses indicate sample means, box limits indicate the 25th and 75th percentiles, whiskers extend 1.5 times the interquartile range from the 25th and 75th percentiles, bars show 95% confidence intervals of the means and outliers are represented by empty dots. Width of the boxes is proportional to the square root of sample size, $n = 26, 28, 29, 12, 30, 24$ sample points.

Genome size variation among populations within regions (Table S1) was also not significantly different, as indicated by very small standard deviations for the intercept and the residual obtained for the random effects ($\text{SD}_{\text{intercept}} = 0.024$; $\text{SD}_{\text{residual}} = 0.063$).

## DISCUSSION

We found no traces of polyploidization events in the *C. solstitialis* populations investigated and geographic differences in genome size were negligible.

A previous record of isolated tetraploids (one accession) in Northern Turkey (*Inceer, Hayirlioglu-Ayaz & Ozcan, 2007*) is intriguing, since further genomic sampling in the area (e.g., less than 40 km from the initial site, *Barker et al., 2017*) did not validate the findings. Further investigation is also required to clarify the reported putative hybridization (*Barker et al., 2017*) with *Centaurea nicaeensis L* . (2n = 20 *chromosomes, Guinochet & Foissac, 1962), since* inter-specific hybridization does not seem to have played a significant role in the past invasion history of *C. solstitialis* (*Barker et al., 2017*). Formerly, a single natural hybrid of *Centaurea ×moncktonii* CE Britton and *C. solstitialis* was described from Oregon, USA (*Roché & Susanna, 2010*) and found to be a sterile triploid (*Miskella, 2014*).

The genome size value we obtained for California (1.72 pg/2C, SD = 0.07 pg) was similar to the one previously reported for Southwestern Oregon (1.66 pg/2C, SD = 0.07 pg), by *Miskella (2014)* and, overall, genome sizes were similar among the six world regions.

**Table 1** Genome size estimations in *Centaurea solstitialis* across the six sampled regions.

| Region | Genome size (2C, pg) | | | | | N |
|---|---|---|---|---|---|---|
| | Mean | SD | SE | Min | Max | |
| Argentina | 1.727 | 0.067 | 0.012 | 1.53 | 1.84 | 29 |
| Australia | 1.705 | 0.061 | 0.012 | 1.59 | 1.83 | 24 |
| California | 1.727 | 0.074 | 0.013 | 1.59 | 1.85 | 30 |
| Chile | 1.717 | 0.065 | 0.018 | 1.59 | 1.81 | 12 |
| Spain | 1.709 | 0.069 | 0.013 | 1.57 | 1.83 | 28 |
| Turkey | 1.737 | 0.070 | 0.013 | 1.60 | 1.88 | 26 |
| **Total** | **1.720** | **0.068** | **0.014** | **1.57** | **1.84** | **149** |

**Notes.**

Values are given as mean, standard deviation and standard error of the mean. The minimum and maximum values and the number of analyzed individuals (*N*) for genome size estimations are also provided.

In conclusion, our thorough sampling of the most representative native and non-native populations across the world's distribution of *C. solstitialis* indicates that its invasive success is not due to changes in genome size or ploidy level. We cannot discard that some individuals in some unsampled populations could present some degree of polyploidy, but their role in invasive success, to date, would have been of minor importance.

# ACKNOWLEDGEMENTS

We are grateful to Joan Vallès (Barcelona) and three other anonymous reviewers for their valuable comments on the previous version of this manuscript.

## Funding

This study was funded by the Portuguese Fundação para a Ciência e a Tecnologia (FCT) of the Ministério da Ciência, Tecnologia e Ensino Superior, with national funds PTDC/BIA-PLA/0763/2014. The funders had no role in study design, data collection and analysis, decision to publish, or preparation of the manuscript.

## Grant Disclosures

The following grant information was disclosed by the authors:
Portuguese Fundação para a Ciência e a Tecnologia (FCT) of the Ministério da Ciência, Tecnologia e Ensino Superior: PTDC/BIA-PLA/0763/2014.

## Competing Interests

Christopher J. Lortie is an Academic Editor for PeerJ. The authors declare there are no competing interests.

## Author Contributions

- Ramona-Elena Irimia and João Loureiro conceived and designed the experiments, performed the experiments, analyzed the data, contributed reagents/materials/analysis tools, wrote the paper, prepared figures and/or tables, reviewed drafts of the paper.

- Daniel Montesinos conceived and designed the experiments, analyzed the data, contributed reagents/materials/analysis tools, wrote the paper, prepared figures and/or tables, reviewed drafts of the paper, collected plant seeds.
- Özkan Eren, Christopher J. Lortie, Kristine French, Lohengrin A. Cavieres, Gastón J. Sotes and José L. Hierro reviewed drafts of the paper, collected plant seeds.
- Andreia Jorge performed the experiments, reviewed drafts of the paper.

### Data Availability

The raw data is available as a Supplementary File.

### Supplemental Information

Supplemental information for this article can be found online at http://dx.doi.org/10.7717/peerj.3531#supplemental-information.

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
