# Peer review of "Extensive analysis of native and non-native Centaurea solstitialis L. populations across the world shows no traces of polyploidization"

_PeerJ, doi:10.7717/peerj.3531_

## Round 0.1 · original submission · Minor Revisions

The reviewers are in general agreement about the soundness of the findings and the merit of the work, but do offer some valuable suggestions for improvement. I wish to give you the opportunity to respond to these at your own discretion, and ask that you provide an item-by-item rebuttal detailing what you have chosen to modify (or not).

Reviewer 1 ·

Basic reporting

English is clear throughout.
Literature is sufficient.
Structure of paper is appropriate.

The rationale for undertaking this work was that polyploidy may be an important factor in the invasion history of this species. That could be the relevant hypothesis. The study refutes this hypothesis, which is informative. Because what I believe was the relevant hypothesis was refuted, the authors seem to feel obligated to erect a new hypothesis that the success of the species is due to its small genome. This does not seem to be a good strategy.

Experimental design

Experimental design and methods are all reasonable and rigorously performed.

Validity of the findings

Findings are valid. Replication is extensive and data is robust and sound.

Additional comments

The manuscript by Irimia et al. characterizes the genome size and thus polyploid levels of Centaurea solstitialis from populations in both native and invasive regions of its range. The methods and analyses are appropriate and the data are presented in a reasonable manner. The paper confirms that this species is diploid in both native and invasive regions; a notion previously reported, but not thoroughly investigated.

The manuscript is well written and brief as it should be. I have only a few minor criticisms that the authors should consider. My criticisms are not at all with the data and their relevance. The criticisms are all focused on what seems to be a prevailing need/intent by the authors to explain why there is no polyploidy and still to use genome size as an explanation for the success of this species as an invader. The comments are below.

Lines 32 and 36-37: The introduction is a little odd in that the first line (32) states that ploidy changes (increases in genome size) are important for invasion success, while the next thought in line 36-37 states that small genomes are important for invasion success. The paragraph continues and gives examples of successful small genome invaders and large genome invaders. Further, in line 44, they mention the study which noted that ¼ of the worst invasive plant species have multiple ploidy levels, or conversely, ¾ don’t have genome size changes (is this proportion any different from other groups of plants?). Together this approach in balance, seems to suggest that the genome size is irrelevant. Of course it is not, and I believe that the authors are just emphasizing that evolution works in different ways and that there are multiple factors that may lead to, or drive, the success of a species; genome size changes might not be the only one, or even the critical one. The authors can certainly make the point that polyploidy does seem to have been an important factor in the evolution of some invasive species, but for many species, including C. solstitialis, it has not been important. They could make this point a little clearer.

Line 138-139: The authors state that it is interesting that Australian populations have the smallest genome size. Why is this interesting? Given that there was no polyploidy, it feels like the authors are trying to make a case now that a small genome in this species is important for invasion success (see above comment). Given that they sampled 2 native and 4 invasive regions, and that one has to be the smallest, that distinction would most likely land on an invasive site completely by chance, in this case Australia. The point is that there is no native/invasive pattern here and the values are all the same anyway. I think it would be good to avoid forcing some significance into this.

Lines 143-146: As in the comment above, the authors seem to be trying to explain something that did not happen or add meaning to a non-event. They seem to imply that admixture and something called “the mutational factor” prevent polyploidy, and that polyploidy would only evolve in introduced places. I don’t believe any of those are true. Also, I do not understand what is meant by “…strong enough factor to trigger such a complex process as polyploidization.” Breakdown in the cell cycle and somatic chromosome doubling (polyploidy) does not seem so complex and happens frequently in many organisms. What drives the success of lineage initiated by one of these errors is of course not so obvious and likely very complex.

Lines 155-157: Once again, since there was no polyploidy or increased genome size detected, the authors seem to be trying to lay the invasion success of this species on a small genome. This line 155-157, is deceptive in that they state that C. solstitialis is among the smallest “diploid” genomes in the genus. The range they give for these diploids is 0.74-3.46 pg with C. solstitialis at 0.87. I did not look at all the original data, but I did look at the Kew C-value database which lists the studies cited in this manuscript, and the 1C range I see there for the likely true diploids is only as high as 1.68 with 2/3 of those under 1.1 pg. Those taxa with higher values are defined as tetraploids; they are diploidized polyploids in many cases. So, yes, C. solstitialis is on the smaller end of the distribution, but it is not particularly noteworthy in any specific way.

In summary, the data are good, useful and should be published, but the authors might want to tone down the emphasis that genome size is relevant in the discussion of this species’ invasion history.

·

Basic reporting

The manuscript is scientifically sound, clear and easy to follow. See more details in the general comments.

Experimental design

The manuscript is composed of original primary research. See more details in the general comments.

Validity of the findings

The findings are significant and robust. See more details in the general comments.

Additional comments

This manuscript reports a big amount of data on genome size on a high number of populations and individuals of Centaurea solstitialis covering its entire distribution area as native or non-native. The work is well designed, the techniques seem to have been adequately used, and the results are numerous and interesting, and are discussed with the help of pertinent literature. I believe the manuscript is an elegant example of the use of a large number of genome size data to establish ploidy levels and to establish whether a case of plant expansion is due or not to polyploidy. I find the work fully worthy of publication in PeerJ, and I have just a few suggestions to improve it, which I develop now.
1.- The herbarium/herbaria where the vouchers have been deposited should be quoted, with indication of its/their international acronym/s, if any.
2.- In case this is in the journal’s rules, the authorities of each taxon should be added in the text the first time a plant name appears.
3.- Use a non-italicised multiplication symbol (×) instead of a letter x, without space between it and the specific epithet, in hybrid names.
4.- All 2C values reported in the manuscript are higher than one of the previous data on the species and lower than another one. Maybe this would deserve a comment. In any case, the authors emphasise that the studied species has a small genome, so that, once polyploidy is discarded, a comment on this fact as facilitator of expansive (and so invasive) character would be good too.
5.- I believe that “woody plant buffer” should not be written with capital initials.
6.- A supplementary table with the results (including coefficients of variation) on each population would be a very valuable complement to the Table S2’mean data.
7.- Although it is sometimes used as singular, data is a plural word (singular: datum), so that I recommend writing “data were” rather than “data was”.

Reviewer 3 ·

Basic reporting

1. The writing quality could be improved. In numerous instances, the wording is awkward or vague.

Examples:
a. Lines 37-39 Here, it sounds like the genome, rather than the organism is doing the invading.
"Several studies support the hypothesis that a smaller genome holds a higher potential to invade because it can boost plant growth and enhance competitive advantage (Bennett et al., 1998; Grotkopp et al.,2004; Beaulieu et al., 2006; Lavergne et al., 2010; Suda et al., 2015)"

b. Lines 72-73. Here, use of the phase "the doubt" makes the sentence a bit awkward, please reconsider rewarding. "Consequently, the doubt remained if ploidy could have played a role in at least some of the C. solstitialis invaded ranges."

2. In the Introduction, the authors discuss how polyploidization may both increase and limit the potential of species to become invasive. Their Introduction would benefit from a more thorough discussion (including examples) of how polyploidization may increase invasive potential, rather than simply pointing the readers to a review (the te Beest et al. paper).

3. The authors say that C. solstitialis is economically important (lines 53-56), but do not explain how. More references and details are needed in order for readers to understand why the work is justified.

Experimental design

1. For the one-way ANOVA, the F-statistic, degrees of freedom, and p-value should be reported.

2. The authors want to determine whether polyploidization has possibly contributed to the invasiveness of C. solstitialis. Although several studies suggested that C. sostitialis was a diploid, there was some evidence of variation in chromosome number and polyploidy in the native range. They sampled a large number of C. solstitialis populations via flow cytometry to determine the ploidy of C. solstitialis across its native and introduced range. Again, with more references and discussion of the problems associated with this invasive species (see point 3 above), the research would be well justified.

Validity of the findings

No comment -- the flow cytometry data presented is entirely unambiguous and the experiments were conducted appropriately.

Additional comments

The use of the word "claimed" in the following sentence (lines 70-72) implies doubt, which doesn't appear to be warranted. Inceer et al. performed chromosome counts, which are a highly reliable method of determining ploidy.

"Furthermore, Inceer et al. (2007) claimed to have found tetraploids in seeds sampled in northern Turkey, but none of those observations have been confirmed since then."

Line 94 "Flow Cytometer" should be "Flow Cytometry"

Reviewer 4 ·

Basic reporting

The basic reporting is fine. I make a couple suggestions to try to improve clarity, but generally the English is clear, unambiguous, and professional. I suggest a couple additional references, and positions where other references should be included, but generally the literature and structure is good. The manuscript is also self-contained with relevant results.

Experimental design

The basic experimental design is sufficient. I am not familiar with flow cytometry, so cannot comment on the experimental design related to that method. The sampling appears sufficient, and I assume that the low number of samples from each population to estimate genome size is due to cost restrictions to run the flow cytometer. There are also quite a few populations, and 3 individuals from each population should be sufficient sampling. I do, however, have some questions regarding the lumping of populations into countries which are addressed in the notes to the authors. The research appears to be within the aims and scope of the journal, are well-defined and meaningful, and it states about the research fills an identified knowledge gap. I'm not entirely convinced that knowledge gap really existed, as sufficient evidence from other sequencing efforts all indicated Centaurea solstitialis was diploid, but this research is important to prove without a shadow of doubt that the species is, in fact, diploid, and therefore represents a valuable contribution to the field. There appears to be have been a rigorous investigation performed to high standards, and the methods are sufficiently described.

Validity of the findings

The validity of the findings appears sound. The data are robust and controlled, the conclusions are well stated and limited to the results. There is some speculation in the first paragraph of the discussion section that could be developed further.

Additional comments

The manuscript confirms that Yellow starthistle is diploid using flow cytometry. I’m not familiar with the methods of flow cytometry, and unfortunately cannot comment on their methods. The manuscript is largely straightforward, and though not earth-shattering, is an important addition to the literature on yellow starthistle. I suggest some wording changes for clarity, but my single (minor) concern is the lumping of populations in the statistics. They find no statistical differences among groups of populations (Australia, Argentina, Chile, etc), but they don’t report statistics for differences among populations. Countries are largely artificial boundaries, and there may or may not be any biological significance to them. Two populations from Chile and Argentina may or may not be more connected by migration than two populations from California. It becomes cumbersome, however, to report the statistics for all 51 populations, and the sample size for genome size within each population is small. So it may be more informative to present the results as done here. I’d like to see an ANOVA of genome size including all 51 populations, using a strict alpha value for significance. If there are very significant differences among populations, the authors may decide to re-work the figure. They may also decide that the figure is too complicated with all 51 populations, even if there are significant differences. If there are no significant differences among populations (seems likely), they will probably decide to keep the figure and supplemental table 2 as is. The authors might also consider adding the average and SD of genome size per population to supplementary table 1. The main gist of the paper, however, is that flow cytometry confirms that Ceso is diploid, and that conclusion is not affected by this comment.
Line 9: Suggest adding the word “plant” to the first line of the abstract. It is clear to those of us familiar with Ceso that it’s a plant, but perhaps not to others.
Line 12: Suggest replacing “being” with “as they are” – the sentence would read: “. . . . numerous plant biological invasions, as they are involved in trait shifts that are critical to invasive success.” Or something to that effect. The word “being” is awkward in that context.
Line 14: Please replace “data was scarce” with “data are scarce.”
Line 15: Suggest removing “here”
Line 44-45: Out of curiosity, what is the percentage of plant species in general that have multiple ploidy levels? A quarter of invasive species with multiple ploidy levels doesn’t seem like a lot, and could be no different than the total number of plant species with multiple ploidy levels. This question is beyond the scope of this paper, however, and does not need to be addressed for this review.
Line 52: Please replace “less” with “fewer”
Line 57: Dlugosh et al. 2015 and Eriksen et al. 2012 also reported differences in size in native and non-native ranges. Eriksen et al. is already cited elsewhere, but the Dlugosh reference is this:
Dlugosch, K.M., Cang, F.A., Barker, B.S., Andonian, K., Swope, S.M. and Rieseberg, L.H., 2015. Evolution of invasiveness through increased resource use in a vacant niche. Nature plants, 1.
Line 63-73: Doesn’t Dlugosh et al. 2013 mention that their sequencing efforts suggest Ceso is diploid?
Line 72: Please remove “the” – sentence should read: “Consequently, doubt remained if ploidy . . .”
Line 82-84: Sentence might be more clear if it read: “Within the native area, we sampled ten populations from Turkey, near the Caucasus region, where high genetic diversity has been detected and thus is regarded as the site of origin of yellow starthistle. . . .” Also, doesn’t Dlugosh et al. 2013 also regard this as the center of diversity?
Fig 1: Consider replacing the x-axis label “genotype” with “Population Origin” or something to that effect. You may also consider adding a second x-axis label denoting which are native and non-native countries.
Lines 135-139: As per the first paragraph of this review, the populations grouped by country are artificial groupings and may or may not reflect actual biological populations. Depending on whether there are statistically significant differences in genome size among populations within these countries, the authors may decide to revise this section.
Lines 142-146: In general, this paragraph seems very weak and speculative. Short introduction dates are probably irrelevant, because most invasive plants are more recent introductions, and the quarter or so that have multiple ploidy levels would also fall under that category. A short review of the causes of polyploidization may be appropriate here to back up the speculation. An additional hypothesis as to why Ceso has not developed different ploidy levels may involve reproduction strategies. As an obligate outcrosser, the ability of a spontaneously mutated polyploid to reproduce would be extremely limited. But it should be worth noting that you probably don’t need to speculate why Ceso hasn’t developed different ploidy levels at all, since most plants haven't developed different ploidy levels either.

---

## Round 0.2 · accepted · Accept

Thank you for your detailed response to the reviews. Their recommendations have been satisfactorily addressed. I have just a few minor issues that I would recommend you address during production.

line 42. "Although there is still ongoing debate about the prevalence of polyploids among plant species, it has been recently inferred that polyploidy has had a key role in the evolution of most angiosperm lineages (Soltis et al., 2009)." This new passage strikes me as a non-sequitor.

line 58 "divergences". a better term would be "differentiation"

Table S1
"WSG84 datum" is better expressed as "WSG84 format"
Please provide units of genome size in the caption.

Table S2
It is not clear how "for ploidy level only" samples appear in the results